# Bandwidth Enhancement of a V-Band Klystron with Stagger-Tuned Multiple Radial Re-Entrant Cavities

**DOI:** 10.3390/s23177471

**Published:** 2023-08-28

**Authors:** M. Santosh Kumar, Santigopal Maity, Soumaya Mandal, Debasish Pal, Chaitali Koley, Ayan Kumar Bandyopadhyay

**Affiliations:** 1National Institute of Technology Mizoram, Aizawl 796012, Indiasantigopal.100@gmail.com (S.M.); s.mandal711312@gmail.com (S.M.); 2CSIR-Central Electronics Engineering Research Institute, Pilani 333031, India; 3Academy of Scientific and Innovative Research (AcSIR), Ghaziabad 201002, India

**Keywords:** V-band Klystron, stagger tuning, bandwidth enhancement, vacuum electronic amplifiers, high data rate communication

## Abstract

The V-band frequencies are becoming popular due to their application potential towards secure high data rate communications. This article reports bandwidth enhancement of an 11-cavity V-band Klystron amplifier employing staggered tuning. A systematic approach is presented to stagger-tune the periodically allocated multiple cavities of the Klystron operating at 60.1 GHz. Using the three-dimensional particle-in-cell (PIC) simulation, it is shown that, employing the proposed approach, the −3 dB bandwidth of the device (with peak tuned configuration) has been increased from 165 MHz to 540 MHz, demonstrating a 260% increment. The −1 dB bandwidth of the device is estimated to be 270 MHz. The proposed approach of stagger tuning may be employed for similar devices employing multiple RF cavities to meet the requirement of wide bandwidth.

## 1. Introduction

In the modern era, the V-band frequency range is increasingly used for high data rate secure communications in various fields like short-range communication in the strategic sector, inter-satellite communication, over-the-air wireless backhaul, etc. [1,2]. Vacuum electronic tube amplifiers, capable of providing high RF output power with excellent efficiency and linearity, play a pivotal role in determining the overall performance of inter-satellite systems and some ground-based high-data-rate systems [3]. To improve the efficiency, bandwidth, and power of vacuum electronic devices, significant research works are being pursued, including various types of Klystrons. In this high-frequency region, particularly for high-bandwidth applications, traveling wave tubes have been used for a long time. However, regarding miniaturization, compactness, and manufacturing simplicity, Klystron amplifiers are also becoming popular choices in the mm-wave region. Klystron amplifiers are popular choices for high-power, phase-stable applications. Recently, these devices have been proposed for potential applications in mm-wave communications, including 5G and beyond applications. The need for ultra-high data rate in wireless connectivity pushes the current 5G mm-wave frequencies up at other millimeter wave bands (30–300 GHz), where abundant bandwidth enables high-speed indoor/hotspot connections [4]. Since the V-band is license-free in most countries, wireless access networks using this band are also popular.

A Klystron amplifier is a linear-beam vacuum electronic device where the key components are an electron gun, a focusing scheme employing magnets, cavity resonators, and input/output windows forming the RF section and collectors. In this amplifier, the accelerated electron beam passes through the input cavity resonator, and the beam’s velocity is modulated according to the input signal. The velocity modulation generates the current modulation in the beam by forming the denser electron bunch as the beam traverses through the successive intermediate cavities. The signal is amplified by the transfer of the kinetic energy of the electron beam when the bunches pass the output cavity, and the cavity fields decelerate their motion. Under this condition, maximum power is extracted from the beam at the gap of the output cavity, and the RF field grows. The gain of this amplifier can be up to 60 dB, with output power up to tens of MW, depending upon the operating frequency and configuration of the device. Usually, the bandwidth of this amplifier can reach up to 10% in some devices [5,6].

The Klystron bandwidth is determined by two main factors: the interaction phenomena between an electron beam and RF waves and the quality factors of the output resonator circuit. Optimization of the interaction phenomena is achieved by the detuning of intermediate resonators and reduction in their Q-factors [7]. The use of multi-cavity interaction structures provides high interaction impedance, resulting in the possibility of efficient beam–wave interaction.

Over an extended period, several design categories for enhancing bandwidth in Klystron have been researched [8,9,10,11,12,13,14,15,16]. Sheet beam type Klystrons, extended interaction type Klystrons, and triaxial Klystrons are popular candidates among them [17,18,19,20,21]. However, the fabrication difficulties limit their use. Using radial re-entrant cavities simplifies Klystron design and fabrication technology and provides higher shunt impedance and R/Q. The detailed design approach of Klystron employing radial re-entrant cavities (in synchronous tuning mode) stems from the article reported in [22]. The radial re-entrant type Klystrons, as reported in [22], are one of the potential candidates for efficient beam–wave interaction.

In contrast to narrowband Klystron, to enhance the bandwidth, the use of more cavities and various tuning methods have been employed [23,24,25,26]. The asynchronous tuning technique is a popular choice to enhance the bandwidth of Klystron tubes. In asynchronous tuning, the cavities are tuned to slightly different frequencies for a fixed-tuned broadband operation. Such tuning techniques are typically called “stagger tuning.” Staggered tuned bunching circuits with input and intermediate cavities can produce a well-bunched electron beam across a wide frequency range.

The staggered tuning method is popularly used for multi-cavity Klystrons, generally employing up to five to six cavities. This paper illustrates the staggered tuning method applied to a V-band Klystron employing 11 cavities. A novel and systematic approach of stagger tuning has been used for a device employing a large number of cavities, reported in [27,28]. The proposed device is simpler in design with a compact form factor and microfabrication compatibility, which are the main advantages of this device. This approach employs extensive three-dimensional simulation of the individual cavities (Eigen-mode simulations) and PIC simulation of the complete RF section at each frequency point of interest. The results of the PIC simulations for each tuning configuration of the device have been presented. Section II describes the design and parameters of the radial re-entrant squared cavities (RRSC). In this section, the design procedure of the RF section is also described, along with a flow chart. Section III focuses on the results of the PIC simulation and also analyzes the performance of the RRSC Klystron. Section IV summarizes the key findings of the present work and provides some suggestions for making further improvements in a potential application area.

## 2. Design Approach

The radio frequency section is a crucial component of a Klystron. The design of the RF section of a multi-cavity Klystron involves the design of intermediate, input, and output resonant cavities (shown in Figure 1) and beam–wave interaction structure. The first cavity with the coupler of the radio frequency section is called the input cavity, or “buncher” cavity, while the last cavity with the coupler of the radio frequency section is called the output cavity, or “catcher” cavity and other cavities are called the intermediate cavities. The structure of the RF section for the proposed Klystron is shown in Figure 2.

Radial re-entrant square cavities (RRSC) have been considered for the multi-cavity Klystron because they are designed to concentrate strong electric fields and are commonly used in high-gain and high-power applications. Two different types of RRSC have been considered for the V-Band RF structure. The model of the input and output cavity with a coupler is shown in Figure 1a, and the model of the intermediate cavity is shown in Figure 1b. A schematic diagram of the designed radial re-entrant square structure intermediate cavity is shown in Figure 1c. In the schematic diagram, the dimensions a, b, c, w, and g represent the height of the cavity, re-entrant height, radius of the beam tunnel, width of the cavity, and beam interaction gap, respectively.

Dimensions of the designed input, output, and intermediate cavities, along with Eigen-mode simulation results, are shown in Table 1. From Table 1, it has been observed that different frequencies have been used for tuning the intermediate cavity and simulation results like R over Q and quality factor (unloaded). The detailed design approach of the Klystron cavities and the drift length between the cavities can be found in reference [28].

Slot coupling is a popular choice for the input and output cavities, which provides several advantages. It offers enhanced coupling efficiency, allowing for an efficient exchange of electromagnetic energy between the cavity gap and external structures. The placement and optimization of coupling slots are important considerations to achieve desired coupling effects in cavity design. To design the input and output cavities, slot coupling has been used. A parametric optimization approach using CST-Microwave Studio [29] has been employed to achieve the desired coupling with an optimum Q-factor. The cross-section views (perspective, top, and front, respectively) of the input and output cavities with the coupler are shown in Figure 3a–c. Details of the coupler dimensions are (Figure 3): coupler heights CH = 0.2 mm and waveguide height WH = 0.3 mm; iris lengths CL = 1 mm, WL1 = 1 mm, and WL2 = 1 mm; and width W = 0.6 mm.

Staggered tuning has been used for increasing the bandwidth of Klystron through beam–wave interaction simulation. The staggered tuning technique involves tuning each stage of a multi-stage amplifier to slightly different frequencies. It provides benefits such as wider bandwidth and sharper pass-band-stop-band transitions.

After the finalization of the cavity geometrical parameter, the RF section was designed using the CST-PIC (particle-in-cell) solver. The primary objectives of PIC simulation are to achieve a sufficiently amplified stable output RF, considering a single frequency input signal for a particular frequency point. If the output signal is not stable, the fifth point of the Klystron design process (Figure 4) is repeated until we obtain a stable RF signal. After obtaining the stable output, the stagger tuning process has been started for enhancement of the bandwidth.

The PIC simulation process of designing the multi-cavity Klystron employing the stagger tuning technique is represented in the flow chart shown in Figure 4.

## 3. Results and Analysis

The RF structure of the V-band Klystron shown in Figure 2 has been modeled and simulated using a CST PIC solver. The PIC simulation has been run for 100 ns with a 0.5 W input RF signal. Using the PIC solver, different parameters of the RF section, like the number of cavities and distance between cavities, have been optimized, and finally, a 27.14 dB gain has been achieved by using 11 numbers of cavities, including the input and output cavities. A combined view of port signals of the beam–wave interaction is shown in Figure 5. Table 3 presents the design parameters of the RF section for the V-band Klystron and the PIC simulation results. Beam–wave interaction has been performed for different frequencies, and we have obtained different gains for different frequencies. For the estimation of bandwidth, the gain of the RF section has been plotted for different frequencies, as shown in Figure 6. After the simulation, we found that the bandwidth is 165 MHz, which is not very significant.

Stagger tuning is one of the popular techniques for operating bandwidth enhancement of Klystron. We have been applying stagger tuning of the intermediate cavities for enhancement of the operating bandwidth by tuning each individual cavity at slightly different frequencies. The RF section of the proposed V-band Klystron has been designed using 11 cavities, including nine numbers of intermediate cavities (IC). The energy diagram is recorded by phase space monitors at different time steps, and the beam trajectory is recorded by a particle monitor along the z-direction, as shown in Figure 7 and Figure 8, respectively. It may be noted that the input RF port is port 1 (marked by “1”) and the output RF port is port 2 (marked by “2”) as shown in Figure 8 while the colour reprsents electron energies. The structure shown in Figure 2 has been designed and simulated using a PIC solver for 100 ns, and the output extracted from the output cavity is shown in Figure 9.

To increase the operating bandwidth, tuning has been started from the last intermediate cavity (IC9) by changing frequency. The frequency tuning has been achieved by changing the transverse dimensions of the particular cavity under tuning. For determining the bandwidth, PIC simulation has been performed by tuning the last intermediate cavity (IC9) from 59.7 GHz to 60.5 GHz with a step range of 100 MHz. The frequencies of other ICs (from IC9 to IC2 and input–output cavities for this instance) were fine-tuned to the earlier configuration, where maximum gain and stability were achieved. Figure 10a shows the bandwidth estimation plot with the optimized frequency of IC9. From Figure 10a, it can be observed that the −3 dB bandwidth is 340MHz. This tuning process has been performed for all other intermediate cavities (IC). Tuning has been performed for IC8, IC7, IC6, IC5, IC4, IC3, IC2, and IC1, respectively. For each and every tuning process, PIC simulation has been performed from 59.7 GHz to 60.5 GHz with a step range of 100 MHz. Frequency vs. gain has been plotted for bandwidth estimation after tuning each of the intermediate cavities.

The details of cavity parameters and tuning frequency of all the cavities of the optimized device with a bandwidth of 540 MHz are mentioned in Table 2.

Figure 10b shows the bandwidth estimation plot after tuning IC8, and it can be observed that the −3 dB bandwidth is 334 MHz. From Figure 10c, the estimated −3 dB bandwidth, after tuning IC7, comes out to be 348 MHz. Similarly, Figure 10d–h show the bandwidth plots of the device after tuning the cavities from IC6 to IC2, which are 347 MHz, 351 MHz, 381 MHz, 372 MHz, and 376 MHz, respectively. Figure 11b shows the bandwidth estimation plot after tuning IC1, and it can be observed that the −3 dB bandwidth is 540 MHz, and Figure 11a shows that the corresponding −1 dB bandwidth is 272 MHz. A moderate bandwidth of 540 MHz has been achieved after tuning the first intermediate cavity, and 260% bandwidth enhancement has been achieved using the stagger tuning process.

The Fourier transform (FT) of the output RF signal is shown in Figure 12. The maximum amplitude of the output signal is achieved at the 60.1 GHz operating frequency. The FT magnitude (normalized to maximum) for the frequency range 59.8 to 60.4 GHz is shown in Figure 12. The bandwidth improvement of the proposed devices is evident from the comparison given in Table 3.

## 4. Conclusions

This article suggests a novel approach of staggered tuning for enhancing bandwidth with an 11-cavity Klystron configuration. In this presented configuration, each of the periodically allocated RF cavities operating at 60.1 GHz (synchronous tuning case) has been systematically tuned for bandwidth enhancement. The design and analysis were performed using a commercial 3D electromagnetic simulation tool. The proposed structure can be fabricated using microfabrication techniques. The proposed approach demonstrates a 260% increment of −3 dB bandwidth (compared to a synchronous tuned configuration) and achieves a −1 dB bandwidth of 270 MHz of the device. Although the achieved bandwidth of the proposed device is less than that offered by traveling wave tubes (TWTs), the proposed device is simpler in design, has a compact form factor, and is compatible with microfabrication, which are the main advantages of this device. The proposed device can be used for wireless high data rate mm-wave communication. The reported approach of staggered tuning of Klystrons employing many cavities may be applied to similar devices operating in other frequencies for their bandwidth enhancement.

## Figures and Tables

**Figure 1 sensors-23-07471-f001:**
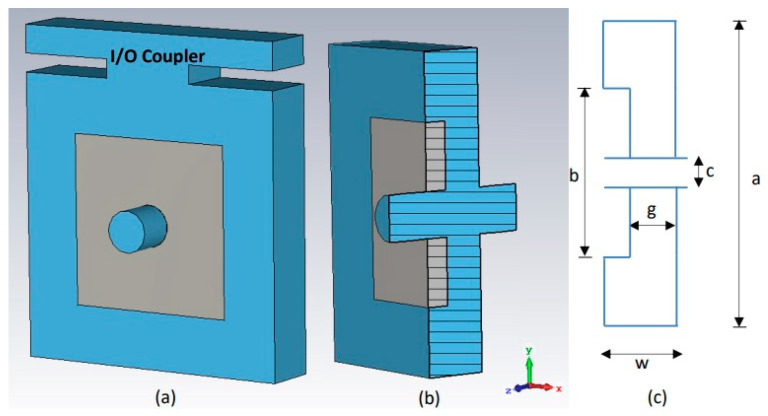
Perspective view of RRSC: (**a**) input/output cavity, (**b**) intermediate cavity, and (**c**) schematic diagram indicating a geometrical parameter of an intermediate cavity.

**Figure 2 sensors-23-07471-f002:**
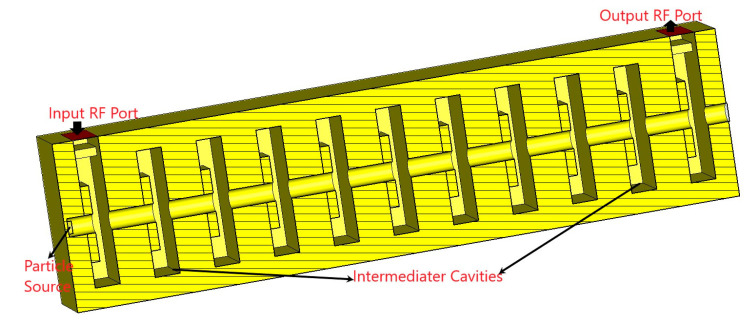
Structure of the RF section for proposed MCK.

**Figure 3 sensors-23-07471-f003:**
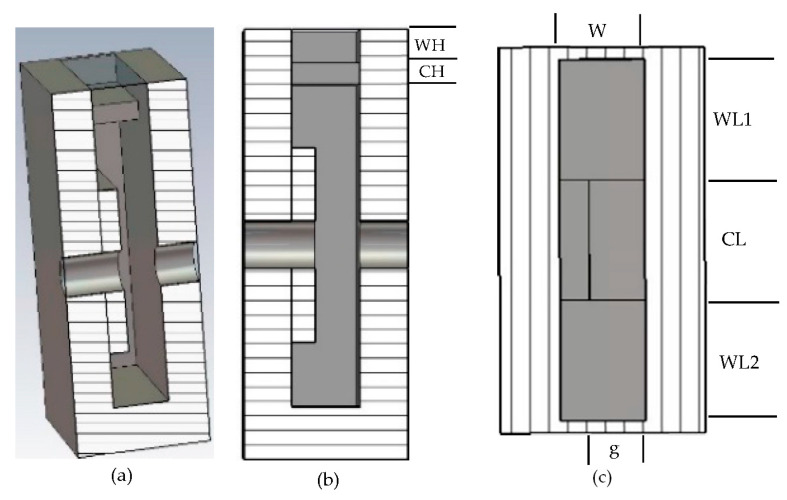
Cross-sectional views of the input–output cavities: (**a**) perspective view, (**b**) side view, and (**c**) top view of the proposed device.

**Figure 4 sensors-23-07471-f004:**
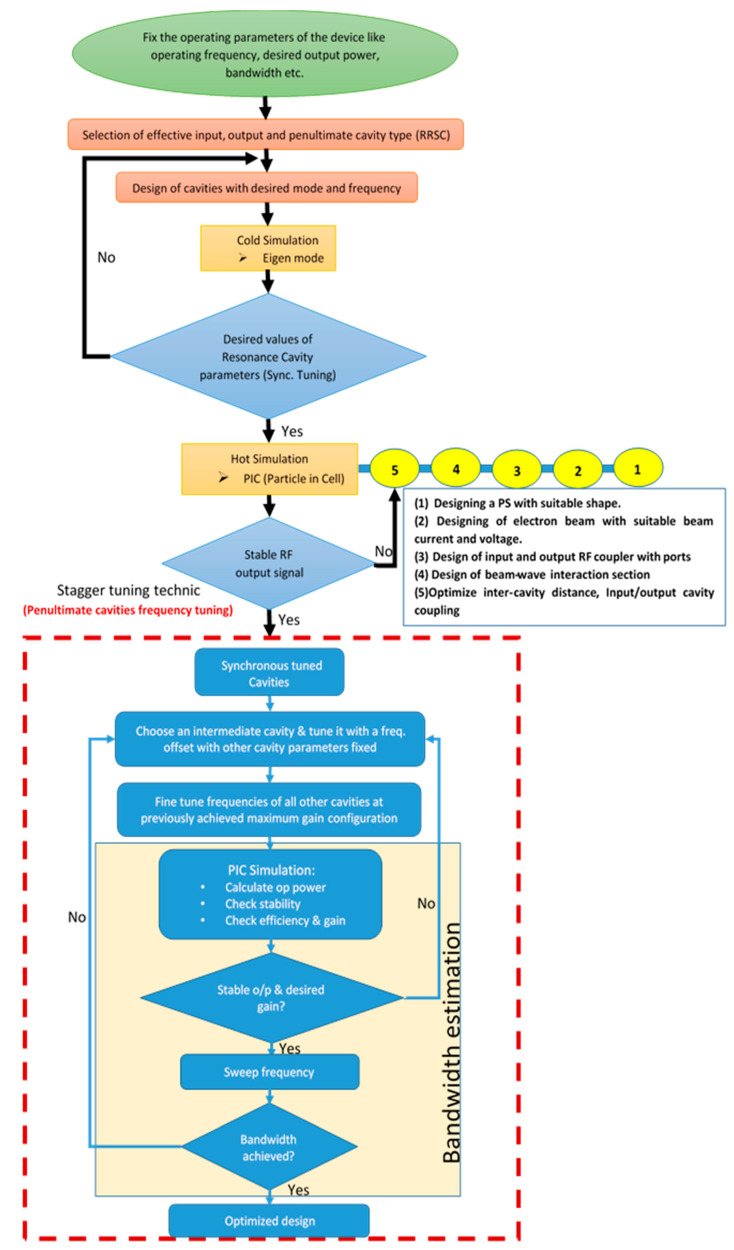
Representation of the stagger tuning technic with a flow chat.

**Figure 5 sensors-23-07471-f005:**
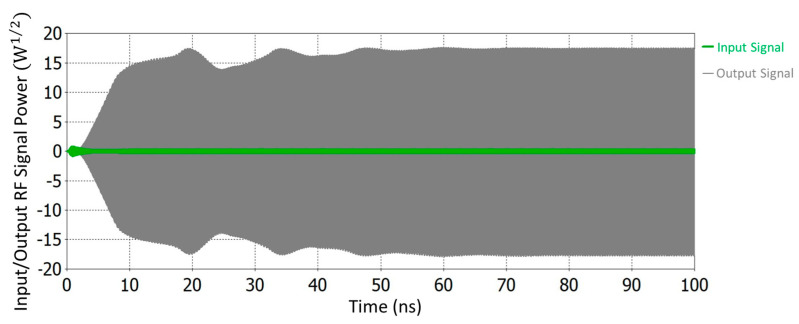
Input signal at input port and amplified RF signal at RF output port (synchronous tuning).

**Figure 6 sensors-23-07471-f006:**
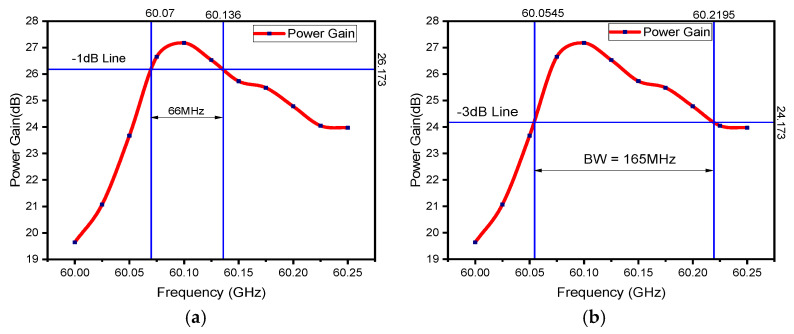
Bandwidth plot of the proposed device in synchronous tuning configuration: (**a**) −1 dB bandwidth plot and (**b**) −3 dB bandwidth plot.

**Figure 7 sensors-23-07471-f007:**
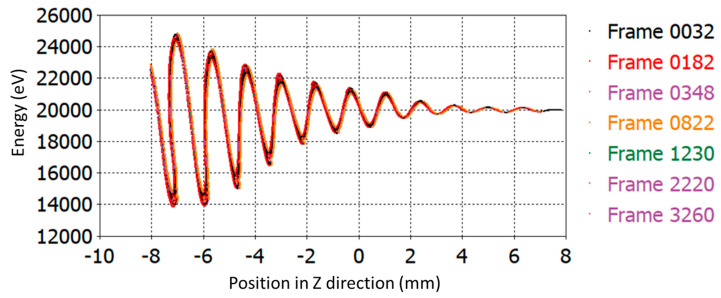
Phase space plot of the particles plotted in z-direction.

**Figure 8 sensors-23-07471-f008:**
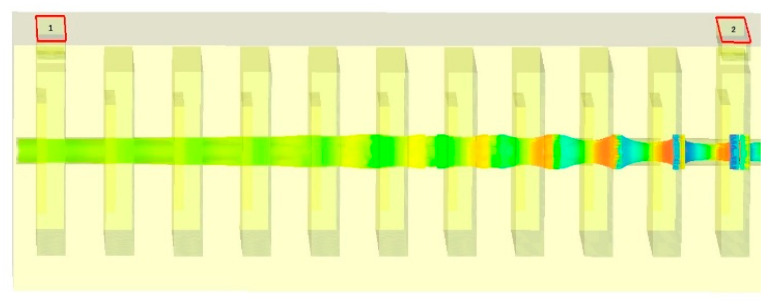
The electron trajectory along the drift tube shows bunching of the same near the output cavity section.

**Figure 9 sensors-23-07471-f009:**
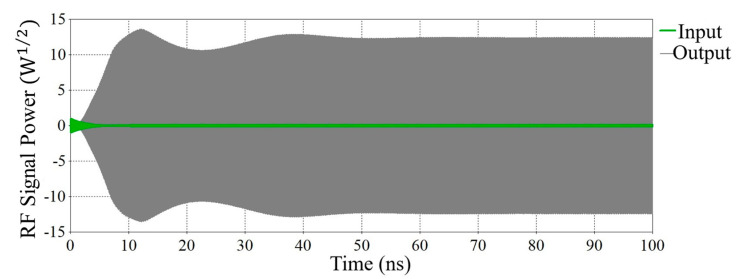
The input signal at the input port and amplified RF signal at the RF output port (stagger tuning).

**Figure 10 sensors-23-07471-f010:**
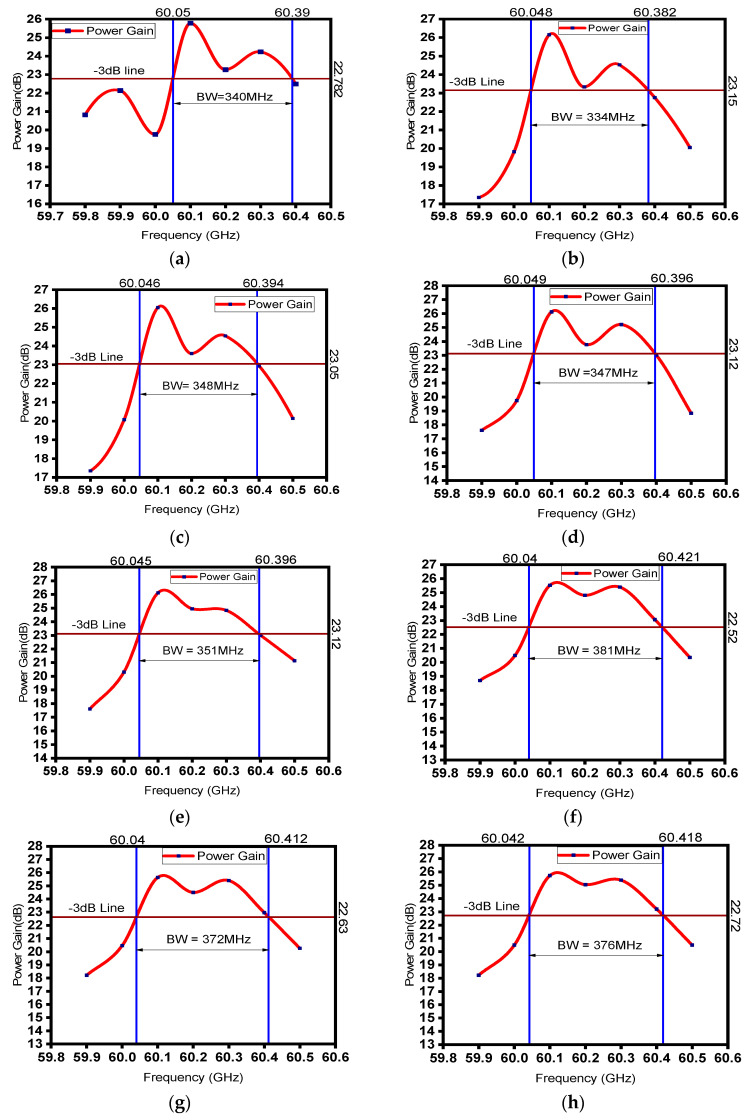
The −3 dB bandwidth plots of the RF section with (**a**) IC9 tuned at 60.1 GHz, (**b**) IC8 tuned at 60.1 GHz, (**c**) IC7 tuned at 60.1 GHz, (**d**) IC6 tuned at 60.1 GHz, (**e**) IC5 tuned at 60.1 GHz, (**f**) IC4 tuned at 60.1 GHz, (**g**) IC3 tuned at 60.1 GHz, and (**h**) IC2 tuned at 60.1 GHz.

**Figure 11 sensors-23-07471-f011:**
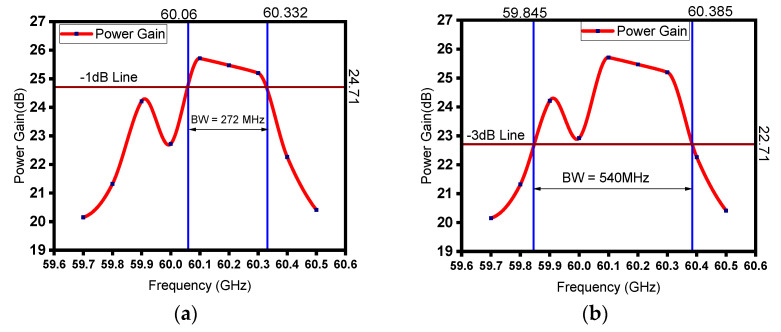
(**a**) The −1 dB bandwidth plot of the RF section with IC1 tuned at 60.4 GHz (all others are tuned at 60.1 GHz) and (**b**) −3 dB bandwidth plot of the same configuration.

**Figure 12 sensors-23-07471-f012:**
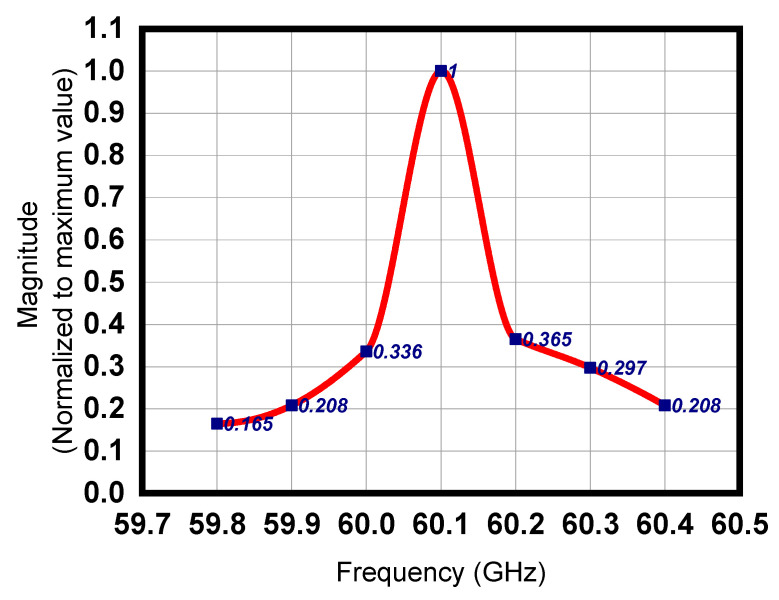
FT magnitude vs. frequency.

**Table 1 sensors-23-07471-t001:** Dimensions of the radial re-entrant square cavities and simulation results.

Cavity Type	Dimensions (mm)					Simulation Results		
	a	w	b	g	c	Q	R/Q (Ω)	f (GHz)
Input–output RRSC	3	0.6	1.8	0.395	0.22	1364	86	60.1
Intermediate RRSC	3	0.6	1.7	0.38	0.22	1359	85	60.1

**Table 2 sensors-23-07471-t002:** Dimensions of all cavities and simulation results of IC1 tuning at maximum bandwidth (540 MHz).

Intermediate Cavity Type	Dimensions (mm)					Simulation Results		
	a	w	b	g	c	Unloaded Q (Q_0_)	R/Q (Ω)	f (GHz)
IC1	3	0.6	1.75	0.3875	0.22	1346	85.7	60.42
IC9	3	0.6	1.6	0.38	0.22	1384	89.2	60.12
IC8	3	0.6	1.6	0.38	0.22	1384	89.2	60.12
IC7	3	0.6	1.6	0.38	0.22	1384	89.2	60.12
IC6	3	0.6	1.6	0.38	0.22	1384	89.2	60.12
IC5	3	0.6	1.6	0.38	0.22	1384	89.2	60.12
IC4	3	0.6	1.6	0.38	0.22	1384	89.2	60.12
IC3	3	0.6	1.6	0.38	0.22	1384	89.2	60.12
IC2	3	0.6	1.6	0.38	0.22	1384	89.2	60.12
Input–Output RRSC	3	0.6	1.8	0.395	0.22	1364	86	60.1

**Table 3 sensors-23-07471-t003:** PIC simulated parameters and results.

Parameter	Synchronous Tuning	Stagger Tuning	Units
Beam Voltage	20	20	KV
Beam Current	200	200	mA
Operating Frequency	60.1	60.1	GHz
Input RF Power	0.5	0.5	W
Output RF Power	256	186	W
Power Gain	27.13	25.61	dBs
−1dB Bandwidth	66	272	MHz
−3 dB Bandwidth	165	540	MHz
Focusing Magnetic Field	0.2	0.2	T
Length of RF Circuit	16	16	mm

## Data Availability

Not applicable.

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
