# Peer review of "Bandwidth Enhancement of a V-Band Klystron with Stagger-Tuned Multiple Radial Re-Entrant Cavities"

_sensors, 2023, doi:10.3390/s23177471_

Round 1

Reviewer 1 Report

The authors presented their work on the wideband klystron at V-band. The improvement achieved by using stagger tunning method has been shown. Here are some comments:

1. What is the advantage of this device? The bandwidth is 540MHz and the output power is 186W. A traveling wave tube operating at this frequency band can offer a much broader bandwidth and a higher output power.

2. The figures need to be redrawn. There should be description for (a) (b) and (c), same problem with Figs. 6 and 11.

3. What is the purpose of figure 4? What is the difference between this procedure and conventional Klystron design?

4. Figs. 5 and 6 are with the same title; same for Figs. 8 and 9.

5. What kind gain were shown in Fig. 9, saturation gain or small signal gain?

6. In Figure 9, what is the resonate frequency of the other cavities when the specified IC tunning frequency is 60.1GHz?

7. Page 10, line 233, "The proposed structure can be fabricated using microfabrication techniques." Indeed, the structure is with a 2D shape, while the ratio g/a reaches 10 in this manuscript, it is very difficult to realize.

No.

Author Response

  1. What is the advantage of this device? The bandwidth is 540MHz and the output power is 186W. A traveling wave tube operating at this frequency band can offer a much broader bandwidth and a higher output power.

The authors fully agree with the fact that a helix or corrugated waveguide-type TWT can offer much wider bandwidth than the proposed klystron. However, the proposed device is simpler in design, with a compact form factor and microfabrication compatibility, which are the main advantages of this device. 

This has been included in the manuscript. (lines 243-245)

  1. The figures need to be redrawn. There should be description for (a) (b) and (c), same problem with Figs. 6 and 11.

These errors have been corrected. (lines 172-173, 197-198)

  1. What is the purpose of figure 4? What is the difference between this procedure and conventional Klystron design?

The conventional klystron, in general, consists of five to six cavities. Here we use periodically allocated re-entrant cavities [details of these types of klystrons can be found in Claudio Paoloni, "Periodically Allocated Reentrant Cavity Klystron," IEEE Trans. Electron Devices, vol. 61, no. 6, pp. 1687–1691, Jun. 2014]. In our case, we have used 11 cavities, including one input and one output cavity. We have shown that by stagger-tuning the cavities of the klystron, we can significantly increase the bandwidth.

The main difference in tuning this type of klystron is the added complexity of stagger-tuning all the cavities (11 in our case instead of 5 or 6 in conventional klystrons). In conventional klystron, the tuning is generally achieved by adding tuners in the cavities; here, we have altered the geometrical dimensions of the cavities for the same.

The systematic procedure of PIC simulation-based stagger tuning has been described in Fig. 4, and the tuning procedure has been explained in the Results Analysis Section.

  1. Figs. 5 and 6 are with the same title; same for Figs. 8 and 9.

These are due to mistakes that have been corrected.

  1. What kind gain were shown in Fig. 9, saturation gain or small signal gain?

It is the saturated gain

  1. In Figure 9, what is the resonate frequency of the other cavities when the specified IC tuning frequency is 60.1GHz?

Yes, the other cavities were tuned at 60.1 GHz. The frequencies of the cavities have been mentioned in a new table (Table 2) in the updated manuscript.  

  1. Page 10, line 233, "The proposed structure can be fabricated using microfabrication techniques." Indeed, the structure is with a 2D shape, while the ratio g/a reaches 10 in this manuscript, it is very difficult to realize.

It will be difficult to realize as one structure. We can think of realizing the same by having it in three parts: the upper and lower parts (plates) and the middle section separately. Another alternate approach is to have it in two halves and combine them using a precise alignment technique.

Reviewer 2 Report

1. Explain clearly novelties of the proposed research with latest works

2. Include the comparison table before references what parameters are good including bandwidth of the design

3.  Add some more latest references around 5 to 8 related to this research work

4. Fig.4 mentioned in the paper not getting clarity, you can improve quality of the figure. Easy to read the readers for better understanding

5. What are the main important parameters for the design of V-band Klystron and how to select that parameters in the design explain clearly to understanding the readers easily

6. 

Need to improve, some grammatical mistakes are there

Author Response

  1. Explain clearly novelties of the proposed research with latest works

Explained in lines 77-80 in the modified manuscript.

  1. Include the comparison table before references what parameters are good including bandwidth of the design

A comparison with an earlier reported device in the same band is given in Table 2 (new table number 3). In the revised manuscript, this table has been placed before the conclusion, as suggested.

  1. Add some more latest references around 5 to 8 related to this research work

The new references have been added to the revised manuscript references (12–16).

  1. Fig.4 mentioned in the paper not getting clarity, you can improve quality of the figure. Easy to read the readers for better understanding

The figure has been changed with better quality for understanding.

  1. What are the main important parameters for the design of V-band Klystron and how to select that parameters in the design explain clearly to understanding the readers easily

The parameters have been given in Table 2. Besides, the selection of the important cavity parameters is also given in another table (new table (Table 2)). 

Reviewer 3 Report

See attached pdf

Please see attached pdf

Author Response

Instead of a conventional stagger tuned klystron, the design structure of this V-band Klystron steams

from C. Paoloni’s “Periodically Allocated Reentrant Cavity Klystron.” ([16] in reference). This is mentioned in the manuscript (line 66), but not clearly stated.

The design of the klystron cavities and drift length between the cavities are not clearly described in this manuscript however, they can be found in reference [22].

The authors need to reference prior literature concerning techniques for broad-banding klystrons. Some examples include:

  1. H. Kreuchen, B.A. Auld and N.E. Dixon, “A Study of the Broadband Frequency Response of the Multicavity Klystron Amplifier,” J. Electronics 2: 529 (1957).
  2. Power Klystrons Today, Michael John Smith, Graham Phillips, Research Studies Press, 1995
  3. George Caryotakis, Chapter 3 in Modern Microwave and Millimeter-Wave Power Electronics

edited by R.J. Barker, J.H. Booske, N.C. Luhmann, Jr., and G.S. Nusinovich

  1. Robert Symon and Rodney Vaughan, Clustered Cavity Klystron
  2. Erling Lien, Wideband tunable klystron amplifier for satellite earth stations, 1974 International Electron Devices Meeting (IEDM)

Response to the comments:

The reference [16] has been referenced properly. (revised manuscript lines 65-67)

The reference of the klystron design article has been clearly mentioned. (revised manuscript lines 118-119)

The suggested references have been added to the revised manuscript. (reference number [12-16] of the revised manuscript)

  1. In the Design Approach sec-on, it is stated in the stagger tuning technique flow chart (Figure 4) that “Fine tune frequencies of all other cavities” is performed after tuning the intermediate cavity with a frequency offset. However, there is no description in the Design or Analysis section of how this is done systematically.

The ambiguity has been corrected by modifying the flowchart and adding the explanation in the text line (205-208).

  1. In the Result Analysis section, the captions in Figure 10 and the description of how each cavity is stagger tuned are ambiguous. In Figure 10, it states that “-3 dB Bandwidth Plots (a) Bandwidth Plot of IC9 tuning Frequency 60.1 GHz…”. Does it mean the Power versus Frequency graph was plotted with IC9 tuned to 60.1 GHz? Likewise for Figure 11.

 The ambiguities have been corrected by changing and clarifying the figure descriptions of Figures 10 and 11. (lines 191-193 and lines 197-198)

In addition, it should be clarified that how the cavities are tuned to particular frequencies (if particular dimension(s) are changed) for readers to understand the process.

Frequency tuning of the cavities has been achieved by altering the geometrical dimensions of the cavities. It has been included in the manuscript. (lines 205-207)

It is also ambiguous to the reader that whether all 9 cavities are detuned to different frequencies

to provide sufficient bandwidth or only IC1 is tuned to 60.4 GHz and all other cavities stay at 60.1

GHz. A table with all the cavity frequencies, unloaded Q value and R/Q of the final design would

provide a clear picture.

A table has been provided as suggested.

Table 2. Dimensions of all cavities and simulation results of IC1 tuning at maximum bandwidth (540 MHz).

Intermediate  Cavity type

   Dimensions (mm)

Simulation results

a

w

b

g

c

Unloaded Q (Q0)

R/Q (Ω)

f (GHz)

IC1

3

0.6

1.75

0.3875

0.22

1346

85.7

60.42

IC9

3

0.6

1.6

0.38

0.22

1384

89.2

60.12

IC8

3

0.6

1.6

0.38

0.22

1384

89.2

60.12

IC7

3

0.6

1.6

0.38

0.22

1384

89.2

60.12

IC6

3

0.6

1.6

0.38

0.22

1384

89.2

60.12

IC5

3

0.6

1.6

0.38

0.22

1384

89.2

60.12

IC4

3

0.6

1.6

0.38

0.22

1384

89.2

60.12

IC3

3

0.6

1.6

0.38

0.22

1384

89.2

60.12

IC2

3

0.6

1.6

0.38

0.22

1384

89.2

60.12

Input-output RRSC

3

0.6

1.8

0.395

0.22

1364

86

60.1

  1. The manuscript lacks sufficient proof reading,

- There are redundant descriptions in multiple places in the manuscript, particularly in the last paragraph of Design Approach section (line 140-144) and Result Analysis section (line 197-218)

- The reference numbers in the manuscript are inconsistent with the Reference section. (line 66, line 78, line 121)

- Typo in Line 163. “bandwidth is 165 GHz” should be 165 MHz.

- Dimension notations “CH” and “WH“ are not labeled clearly in Figure 3, line 125-127.

These errors have been taken care of in the revised manuscript. (lines 146-149, 202-233).

The reference numbers in the manuscript are corrected with the reference section. (lines 66-67, 78-79, 126).

The typing error in line 168 has been corrected.

The dimension notations are labelled clearly in Figure 3, lines 131-132.

The same will also be corrected when preparing the final proof of the article.

Round 2

Reviewer 2 Report

Authors fulfill all the comments